# Flow-based Domain Randomization for Learning and Sequencing Robotic Skills

Aidan Curtis [1]   Eric Li [1]   Michael Noseworthy [1]   Nishad Gothoskar [1]   Sachin Chitta [2]   Hui Li [2]
Leslie Pack Kaelbling [1]   Nicole Carey [2]

## Abstract

Domain randomization in reinforcement learning is an established technique for increasing the robustness of control policies trained in simulation. By randomizing environment properties during training, the learned policy can become robust to uncertainties along the randomized dimensions. While the environment distribution is typically specified by hand, in this paper we investigate automatically discovering a sampling distribution via entropy-regularized reward maximization of a normalizing-flow–based neural sampling distribution. We show that this architecture is more flexible and provides greater robustness than existing approaches that learn simpler, parameterized sampling distributions, as demonstrated in six simulated and one real-world robotics domain. Lastly, we explore how these learned sampling distributions, along with a privileged value function, can be used for out-of-distribution detection in an uncertainty-aware multi-step manipulation planner.

## 1. Introduction

Reinforcement learning (RL) is a powerful tool in robotics because it can be used to learn effective control policies for systems with highly complex dynamics that are difficult to model analytically. Unlike traditional control methods, which rely on precise mathematical models, RL learns directly from simulated or real-world experience (Luo & Li, 2021; Zhu et al., 2020; Schoettler et al., 2020).

However, RL approaches can be inefficient, involving slow, minimally parallelized, and potentially unsafe data-gathering processes when performed in real environments (Kober et al., 2013). Learning in simulation eliminates some of these problems, but introduces new issues in the form of discrepancies between the training and real-world environments (Valassakis et al., 2020).

Successful RL from simulation hence requires efficient and accurate models of both robot and environment during the training process. But even with highly accurate geometric and dynamic simulators, the system can still be only considered partially observable (Kober et al., 2013)—material qualities, inertial properties, perception noise, contact and force sensor noise, manufacturing deviations and tolerances, and imprecision in robot calibration all add uncertainty to the model.

To improve the robustness of learned policies against sim-to-real discrepancies, it is common to employ domain randomization, varying the large set of environmental parameters inherent to a task according to a given underlying distribution (Muratore et al., 2019). In this way, policies are trained to maximize their overall performance over a diverse set of models. These sampling distributions are typically constructed manually with Gaussian or uniform distributions on individual parameters with hand-selected variances and bounds. However, choosing appropriate distributions for each of the domain randomization parameters remains a delicate process (Josifovski et al., 2022); too broad a distribution leads to suboptimal local minima convergence (see Figure 3), while too narrow a distribution leads to poor real-world generalization (Mozifian et al., 2019; Packer et al., 2018). Many existing methods rely on real-world rollouts from hardware experiments to estimate dynamics parameters (Chebotar et al., 2019; Ramos et al., 2019; Muratore et al., 2022). However, for complex tasks with physical parameters that are difficult to efficiently or effectively sample, this data may be time-consuming to produce, or simply unavailable.

An alternative strategy is to learn an environment distribution during training with the aim of finding the broadest possible training distribution that can feasibly be solved in order to maximize the chances of transferring to an unknown target environment. Automating updates to parameter distributions during the training process can remove the need

---

[1]MIT CSAIL [2]Autodesk Research. Correspondence to: Aidan Curtis <curtisa@mit.edu>.

*Proceedings of the $42^{nd}$ International Conference on Machine Learning*, Vancouver, Canada. PMLR 267, 2025. Copyright 2025 by the author(s).

for heuristic tuning and iterative experimentation (Mozifian et al., 2019; OpenAI et al., 2019; Tiboni et al., 2024). In this paper, we present GoFlow, a novel approach for learned domain randomization that combines actor-critic reinforcement learning architectures (Schulman et al., 2017; Haarnoja et al., 2018) with a neural sampling distribution to learn robust policies that generalize to real-world settings. By maximizing the diversity of parameters during sampling, we actively discover environments that are challenging for the current policy but still solvable given enough training.

As proof of concept, we investigate one real-world use case: contact-rich manipulation for assembly. Assembly is a critical area of research for robotics, requiring a diverse set of high-contact interactions which often involve wide force bandwidths and unpredictable dynamic changes. Recently, sim-to-real RL has emerged as a potentially useful strategy for learning robust contact-rich policies without laborious real-world interactions (Noseworthy et al., 2024; Tang et al., 2023b; Zhang et al., 2024). We build on this work by testing our method on the real-world industrial assembly task of gear insertion.

Lastly, we extend this classical gear insertion task to the setting of multi-step decision making under uncertainty and partial observability. As shown in this paper and elsewhere (Tiboni et al., 2024; Mozifian et al., 2019; OpenAI et al., 2019), policies trained in simulation have an upper bound on the environmental uncertainties that they can be conformant to. For example, a visionless robot executing an insertion policy can only tolerate so much in-hand pose error. However, estimates of this uncertainty can be used to inform high level control decisions through task-oriented information gathering (Liang et al., 2020; Curtis et al., 2023). For example, looking closer at objects can result in more accurate pose estimates or tracking objects in the hand to detect slippage. By integrating a probabilistic pose estimation model, we can use the sampling distributions learned with GoFlow as an out-of-distribution detector to determine whether the policy is expected to succeed under its current belief about the world state. If the robot has insufficient information, it can act to deliberately seek the needed information using a simple belief-space planning algorithm.

Our contributions are as follows: We introduce GoFlow, a novel domain randomization method that combines actor-critic reinforcement learning with a learned neural sampling distribution. We show that GoFlow achieves higher domain coverage than fixed and other learning-based solutions to domain randomization on a suite of simulated environments. We demonstrate the efficacy of GoFlow in a real-world contact-rich manipulation task—gear insertion—and extend it to multi-step decision-making under uncertainty. By integrating a probabilistic pose estimation model, we enable the robot to actively gather additional information when needed,

enhancing performance in partially observable settings.

## 2. Related Work

Recent developments in reinforcement learning have demonstrated that policies trained in simulation can effectively transfer to real-world robots, even for contact-rich robotic tasks (Zhang et al., 2024; Tang et al., 2023b; Noseworthy et al., 2024; Jin et al., 2023). A key innovation enabling this transferability is domain randomization (**?**Peng et al., 2017), where environment parameters are sampled from predefined distributions during training, enabling learned policies to generalize to environmental uncertainties upon deployment.

Traditionally, domain randomization requires manually defining these sampling distributions, which can be labor-intensive. To address this, recent work has explored methods to automatically learn these distributions, aiming for maximal generalization with minimal manual effort. One notable approach involves constructing adversarial distributions that challenge the current policy, ensuring coverage of the environment parameter space (Mehta et al., 2020; Wang et al., 2025). While adversarial methods can outperform uniform randomization, they typically assume all environments sampled are solvable, limiting their effectiveness in highly uncertain scenarios.

An alternative line of research adopts a curriculum-based approach, progressively expanding the complexity of the training distribution while maintaining policy success. Specific curricular methods include minimizing divergence from a target distribution using multivariate Gaussians (Mozifian et al., 2019), maximizing entropy through independent beta distributions (Tiboni et al., 2024), and incrementally expanding uniform sampling distributions via boundary sampling (OpenAI et al., 2019). In this work, we propose a novel learned domain randomization technique leveraging normalizing flows (Rezende & Mohamed, 2015) as neural sampling distributions. This approach offers increased flexibility and expressivity over existing parametric methods.

Beyond training robust policies in simulation, learned sampling distributions can be tied to the real-world environmental conditions under which policies are likely to succeed. Previous works have integrated domain randomization with real-world interactions for more informed training distributions (Ramos et al., 2019; Sagawa & Hino, 2025; Ajay et al., 2022; Muratore et al., 2020) or to find the maximally effective real-world strategy (Yu et al., 2018; Ren et al., 2023). However, these methods often necessitate expensive policy retraining or data-intensive evolutionary search based on real-world feedback, posing challenges for real-time applications. Instead, we utilize our learned sampling distribution as an out-of-distribution detector within a multi-step planning framework, enabling fast and data-efficient

information gathering in the real world.

# 3. Background

## 3.1. Markov Decision Process

A Markov Decision Process (MDP) is a mathematical framework for modeling decision-making. Formally, an MDP is defined as a tuple $(\mathcal{S}, \mathcal{A}, P, R, \gamma)$, where $\mathcal{S}$ is the state space, $\mathcal{A}$ is the action space, $P : \mathcal{S} \times \mathcal{A} \times \mathcal{S} \rightarrow [0, 1]$ is the state transition probability function, where $P(s' \mid s, a)$ denotes the probability of transitioning to state $s'$ from state $s$ after taking action $a$, $R : \mathcal{S} \times \mathcal{A} \rightarrow \mathbb{R}$ is the reward function, where $R(s, a)$ denotes the expected immediate reward received after taking action $a$ in state $s$, $\gamma \in [0, 1)$ is the discount factor, representing the importance of future rewards.

A policy $\pi : \mathcal{S} \times \mathcal{A} \rightarrow [0, 1]$ defines a probability distribution over actions given states, where $\pi(a \mid s)$ is the probability of taking action $a$ in state $s$. The goal is to find an optimal policy $\pi^*$ that maximizes the expected cumulative discounted reward.

## 3.2. Domain Randomization

Domain randomization introduces variability into the environment by randomizing certain parameters during training. Let $\Xi$ denote the space of domain randomization parameters, and let $\xi \in \Xi$ be a specific instance of these parameters. Each $\xi$ corresponds to a different environment configuration or dynamics.

We can define a parameterized family of Markov Decision Processes (MDPs) where each $\mathcal{M}_\xi = (\mathcal{S}, \mathcal{A}, P_\xi, R_\xi, \gamma)$ has transition dynamics $P_\xi$ and reward function $R_\xi$ dependent on $\xi$. The agent interacts with environments sampled from a distribution over $\Xi$, typically denoted as $p(\xi)$. [1]

The objective is to learn a policy $\pi : \mathcal{S} \rightarrow \mathcal{A}$ that maximizes the expected return across the distribution environments:

$$J(\pi) = \mathbb{E}_{\xi \sim p(\xi)} \left[ \mathbb{E}_{\tau \sim P_\xi, \pi} \left[ \sum_{t=0}^{\infty} \gamma^t R_\xi(s_t, a_t) \right] \right], \quad (1)$$

where $\tau = \{(s_0, a_0, s_1, a_1, \dots)\}$ denotes a trajectory generated by policy $\pi$ in environment $\xi$. Domain randomization aims to find a policy $\pi^*$ such that: $\pi^* = \arg\max_\pi J(\pi)$.

In deep reinforcement learning, the policy $\pi$ is a neural network parameterized by $\theta$, denoted as $\pi_\theta$. The agent learns the policy parameters $\theta$ through interactions with simulated

---

[1]This problem can also be thought of as a POMDP where the observation space is $\mathcal{S}$ and the state space is a product of $\mathcal{S}$ and $\Xi$ as discussed in (Kwon et al., 2021).

environments sampled from $p(\xi)$. In our implementation, we employ the Proximal Policy Optimization (PPO) algorithm (Schulman et al., 2017), an on-policy policy gradient method that optimizes a stochastic policy while ensuring stable and efficient learning.

To further stabilize training, we pass privileged information about the environment parameters $\xi$ to the critic network. The critic network, parameterized by $\psi$, estimates the state-value function:

$$V_\psi(s_t, \xi) = \mathbb{E}_{\pi_\theta} \left[ \sum_{k=0}^{\infty} \gamma^k r_{t+k} \,\middle|\, s_t, \xi \right], \quad (2)$$

where $s_t$ is the current state, $r_{t+k}$ are future rewards, and $\gamma$ is the discount factor. By incorporating $\xi$, the critic can provide more accurate value estimates with lower variance (Pinto et al., 2018). The actor network $\pi_\theta(a_t | s_t)$ does not have access to $\xi$, ensuring that the policy relies only on observable aspects of the state.

## 3.3. Normalizing Flows

Normalizing flows are a class of generative models that transform a simple base distribution into a complex target distribution using a sequence of invertible, differentiable functions. Let $z \sim p_Z(z)$ be a latent variable from a base distribution (e.g., a standard normal distribution). A normalizing flow defines an invertible transformation $f_\phi : \mathbb{R}^d \rightarrow \mathbb{R}^d$ parameterized by neural network parameters $\phi$, such that $x = f_\phi(z)$, aiming for $x$ to follow the target distribution.

The density of $x$ is computed using the change of variables formula:

$$p_X(x) = p_Z(f_\phi^{-1}(x)) \left| \det\left( \frac{\partial f_\phi^{-1}(x)}{\partial x} \right) \right|. \quad (3)$$

For practical computation, this is often rewritten as:

$$\log p_X(x) = \log p_Z(z) - \log \left| \det\left( \frac{\partial f_\phi(z)}{\partial z} \right) \right|, \quad (4)$$

where $\frac{\partial f_\phi(z)}{\partial z}$ is the Jacobian of $f_\phi$ at $z$. By composing multiple such transformations $f_\phi = f_{\phi_K} \circ \cdots \circ f_{\phi_1}$, each parameterized by neural network parameters $\phi_k$, normalizing flows can model highly complex distributions.

In our work, we employ *neural spline flows* (Durkan et al., 2019), a type of normalizing flow where the invertible transformations are constructed using spline-based functions. Specifically, the parameters $\phi$ represent the coefficients of

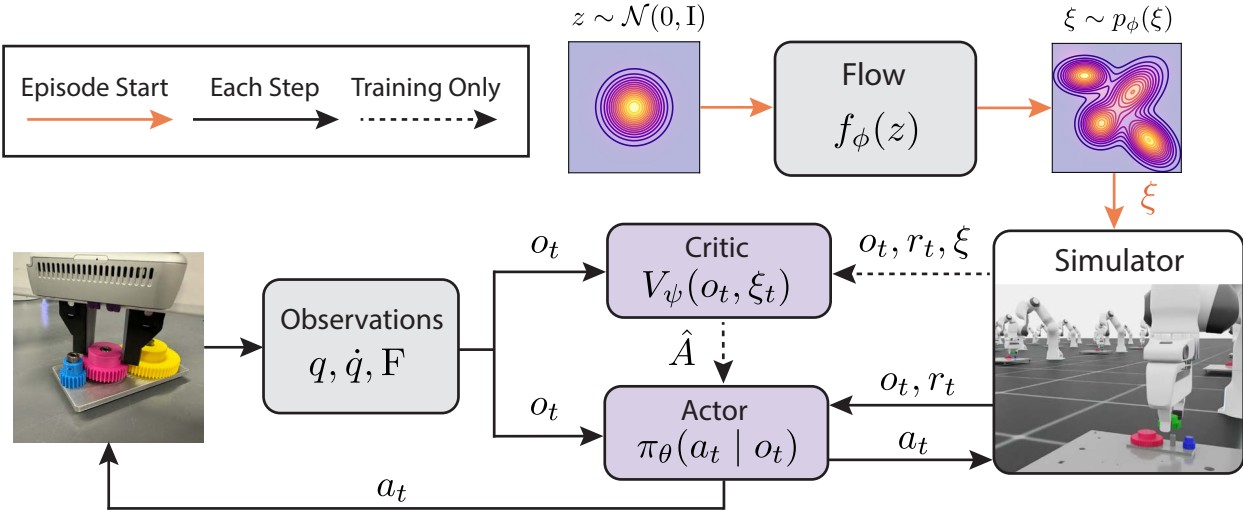

*Figure 1.* An architecture diagram for our actor-critic RL training setup in the gears domain. At the start of each training episode, an environment initialization $\xi \sim p_\phi(\xi)$ is sampled via the normalizing flow model. This variable conditions both the simulator and the privileged critic. At each time step, the agent receives observations including joint positions $q$, joint velocities $\dot{q}$, and force measurements F from which the actor policy generates actions. The critic takes observations $o_t$, reward $r_t$, and environment initialization $\xi$ to estimate the value. This estimated value is used to compute advantage $\hat{A}$ which is then used to train the actor. Only the actor is used during testing and real-world deployment, but the critic and flow can help determine when a policy will be successful as discussed in section 6.

the splines (e.g., knot positions and heights) and the weights and biases of the neural networks that parameterize these splines.

## 4. Method

In this section, we introduce **GoFlow**, a method for learned domain randomization that *goes with the flow* by adaptively adjusting the domain randomization process using normalizing flows.

In traditional domain randomization setups, the distribution $p(\xi)$ is predefined. However, selecting an appropriate $p(\xi)$ is crucial for the policy's performance and generalization. Too broad a sampling distribution and the training focuses on unsolvable environments and falls into local minima. In contrast, too narrow a sampling distribution leads to poor generalization and robustness. Additionally, rapid changes to the sampling distribution can lead to unstable training. To address these challenges, prior works such as (Klink et al., 2021) have proposed a self-paced learner, which starts by mastering a small set of environments, and gradually expands the tasks to solver harder and harder problems while maintaining training stability. This strategy has subsequently been applied to domain randomization in (Mozifian et al., 2019) and (Tiboni et al., 2024), where terms were included for encouraging spread over the sampling

space for greater generalization. We take inspiration from these works to form a joint optimization problem:

$$\max_{p,\pi} \left\{ \mathbb{E}_{\xi \sim p}[J_\xi(\pi)] + \alpha \mathcal{H}(p) - \beta D_{KL}(p_{\text{old}} \| p) \right\} \quad (5)$$

where $p$ is the sampling distribution over environment parameters $p(\xi)$, $\mathcal{H}(p)$ is the differential entropy of $p(\xi)$, $D_{KL}(p \| p_{\text{old}})$ is the divergence between the current and previous sampling distributions, and $\alpha > 0, \beta > 0$ are regularization coefficients that control the trade-off between generalizability, training stability, and the expected reward under the sampling distribution.

Other learned domain randomization approaches propose similar objectives. (Mozifian et al., 2019) maximizes reward but replaces entropy regularization with a KL divergence to a fixed target distribution and omits the self-paced KL term. (Tiboni et al., 2024) includes all three objectives but frames the reward and self-paced KL terms as constraints, maximizing entropy through a nonlinear optimization process that is not easily adaptable to neural sampling distributions. We compare GoFlow to these methods in our experiments to highlight its advantages. To our knowledge, GoFlow is the first method to optimize such an objective with a neural sampling distribution.

The GoFlow algorithm (Algorithm 1) begins by initial-

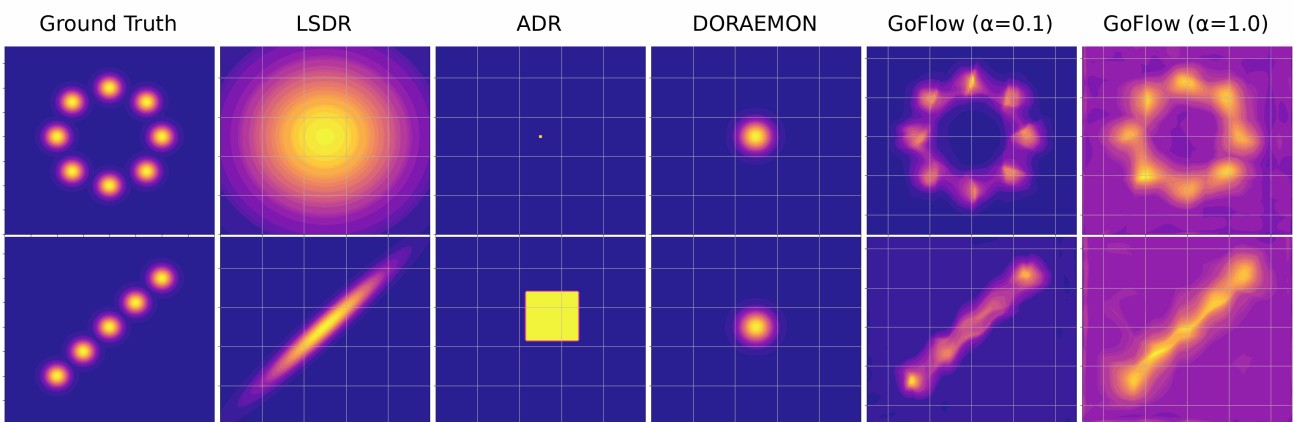

*Figure 2.* An illustrative domain showing the learned sampling functions over the space of unobserved parameters for the tested baselines. Compared to other learning methods, GoFlow correctly models the multimodality and inter-variable dependencies of the underlying reward function. This toy domain, along with other domains in our experiments, violates some of the assumptions made by prior works, such as the feasibility of the center point of the range.

---

**Algorithm 1** GoFlow

---

**Require:** Initial policy parameters $\theta$, flow parameters $\phi$, training steps $N$, network updates $K$, entropy coefficient $\alpha$, similarity coefficient $\beta$, and learning rate $\eta_\phi$

1: **for** $n = 1$ to $N$ **do**
2:     Sample $\{\xi_i^{\text{train}}\}_{i=1}^B \sim p_\phi(\xi)$, $\{\xi_i^{\text{test}}\}_{i=1}^B \sim u(\xi)$
3:     Train $\pi_\theta$ with $\xi_i^{\text{train}}$ initializations
4:     Estimate $J_{\xi_i^{\text{test}}}(\pi_\theta)$ via policy rollouts
5:     Save current flow distribution as $p_{\phi_{\text{old}}}(\xi)$
6:     **for** $k = 1$ to $K$ **do**
7:         $\mathcal{R} \leftarrow \frac{|\Xi|}{B} \sum_{i=1}^B \left[ p_\phi(\xi_i^{\text{test}}) \, J_{\xi_i^{\text{test}}}(\pi_\theta) \right].$
8:         $\hat{\mathcal{H}} \leftarrow -|\Xi| \cdot \mathbb{E}_{\xi \sim u(\xi)} \left[ p_\phi(\xi) \log p_\phi(\xi) \right]$
9:         $\hat{D}_{KL} \leftarrow \mathbb{E}_{\xi \sim p_{\phi_{\text{old}}}(\xi)} \left[ \log p_{\phi_{\text{old}}}(\xi) - \log p_\phi(\xi) \right]$
10:        $\phi \leftarrow \phi + \eta_\phi \nabla_\phi \left( \mathcal{R} + \alpha \hat{H} - \beta \hat{D}_{KL} \right)$
11:    **end for**
12: **end for**

---

izing both the policy parameters $\theta$ and the normalizing flow parameters $\phi$. In each training iteration, GoFlow first samples a batch of environment parameters $\{\xi_i\}_{i=1}^B$ from the current distribution modeled by the normalizing flow (Line 2). These sampled parameters are used to train the policy $\pi_\theta$ (Line 3). Following the policy update, expected returns $J_{\xi_i}(\pi_\theta)$ are estimated for each sampled environment through policy rollouts, providing a measure of the policy's performance under a target uniform distribution $u(\xi)$ after being trained on $p(\xi)$ (Line 4).

After sampling these rollouts, GoFlow then performs $K$ steps of optimization on the sampling distribution. GoFlow

first estimates the policy's performance on the sampling distribution using the previously sampled rollout trajectories (Line 7). The entropy of the sampling distribution (Line 8) and divergence from the previous sampling distribution (Line 9) are estimated using newly drawn samples. Importantly, we compute the reward and entropy terms by importance sampling from a uniform distribution rather than from the flow itself. This broad coverage helps prevent the learned distribution from collapsing around a small region of parameter space. Derivations for these equations can be found in Appendix A.2. These terms are combined to form a loss, which is differentiated to update the parameters of the sampling distribution (Line 10). An architecture diagram for this approach can be seen in Figure 1.

## 5. Domain Randomization Experiments

Our simulated experiments compare policy robustness in a range of domains. For full details on the randomization parameters and bounds, see Appendix A.3.

### 5.1. Domains

First, we examine the application of GoFlow to an illustrative 2D domain that is multimodal and contains intervariable dependencies. The state and action space are in $\mathbb{R}^2$. The agent is initialized randomly in a bounded x, y plane. An energy function is defined by a composition of Gaussians placed in a regular circular or linear array. The agent can observe its position with Gaussian noise proportional to the inverse of the energy function. The agent is rewarded for guessing its location, but is incapable of moving. This task is infeasible when the agent is sufficiently far from any

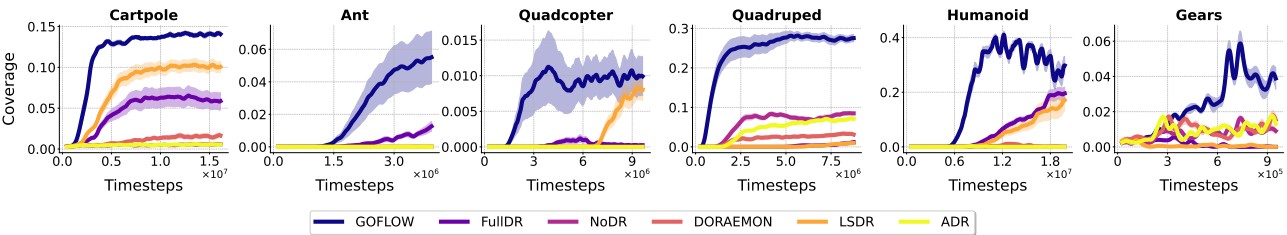

*Figure 3.* The coverage ratio over the target distribution across five random seeds for each of the environments. The bands around each curve indicate the standard error.

of the Gaussian centers, so a sampling distribution should come to resemble the energy function. Some example functions learned by GoFlow and baselines from Section 5.2 can be seen in Figure 2.

Second, we quantitatively compare GoFlow to existing baselines including Cartpole, Ant, Quadcopter, Quadruped, and Humanoid in the IsaacLab suite of environments (Mittal et al., 2023). We randomize over parameters such as link masses, joint frictions, and material properties.

Lastly, we evaluate our method on a contact-rich robot manipulation task of gear insertion, a particularly relevant problem for robotic assembly. In the gears domain, we randomize over the relative pose between the gripper and the held gears along three degrees of freedom. The problem is made difficult by the uncertainty the robot has about the precise location of the gear relative to the hand. The agent must learn to rely on signals of proprioception and force feedback to guide the gear into the gear shaft. The action space consists of end-effector pose offsets along three translational degrees of freedom and one rotational degree around the z dimension. The observation space consists of a history of the ten previous end effector poses and velocities estimated via finite differencing. In addition to simulated experiments, our trained policies are tested on a Franka Emika robot using `IndustReal` library built on the `frankapy` toolkit (Tang et al., 2023a; Zhang et al., 2020).

### 5.2. Baselines

In our domain randomization experiments, we compare to a number of standard RL baselines and learning-based approaches from the literature. In our quantitative experiments, success is determined by the sampled environment passing a certain performance threshold $J_T$ that was selected for each environment. We measure task performance in terms of *coverage*, which is defined as the proportion of the total sampling distribution for which the policy receives higher than $J_t$ reward. Coverage is estimated via policy rollouts on 4096 uniformly selected environment initialization samples from the total range of parameters. All baselines are trained with

an identical neural architecture and PPO implementation. The success thresholds along with other hyperparameters are in Appendix A.5.

We evaluate on the following baselines. First, we compare to no domain randomization (**NoDR**) which trains on a fixed environment parameter at the centroid of the parameter space. Next, we compare to a full domain randomization (**FullDR**) which samples uniformly across the domain within the boundaries during training. In addition to these fixed randomization methods, we evaluate against some other learning-based solutions from the literature: **ADR** (OpenAI et al., 2019) learns uniform intervals that expand over time via boundary sampling. It starts by occupying an initial percentage of the domain and performs "boundary sampling" during training with some probability. The rewards attained from boundary sampling are compared to thresholds that determine if the boundary should be expanded or contracted. **LSDR** (Mozifian et al., 2019) learns a multivariate gaussian sampling distribution using reward maximization with a KL divergence regularization term weighted by an $\alpha$ hyperparameter. Lastly, **DORAEMON** (Tiboni et al., 2024) learns independent beta distributions for each dimension of the domain, using a maximum entropy objective constrained by an estimated success rate.

We compare coverage across environments sampled from a uniform testing distribution within the environment bounds. Our findings in Figure 3 show that GoFlow matches or outperforms baselines across all domains. We find that our method performs particularly well in comparison to other learned baselines when the simpler or more feasible regions of the domain are off-center, irregularly shaped, and have inter-parameter dependencies such as those seen in Figure 2. We intentionally chose large intervals with low coverage to demonstrate this capability, and we perform a full study of how baselines degrade significantly with increased parameter ranges while GoFlow degrades more gracefully (see Appendix A.6). Additionally, coverage should not be mistaken for success-rate, as these policies can be integrated into a planning framework that avoids the use of infeasible actions or gathers information to increase coverage as

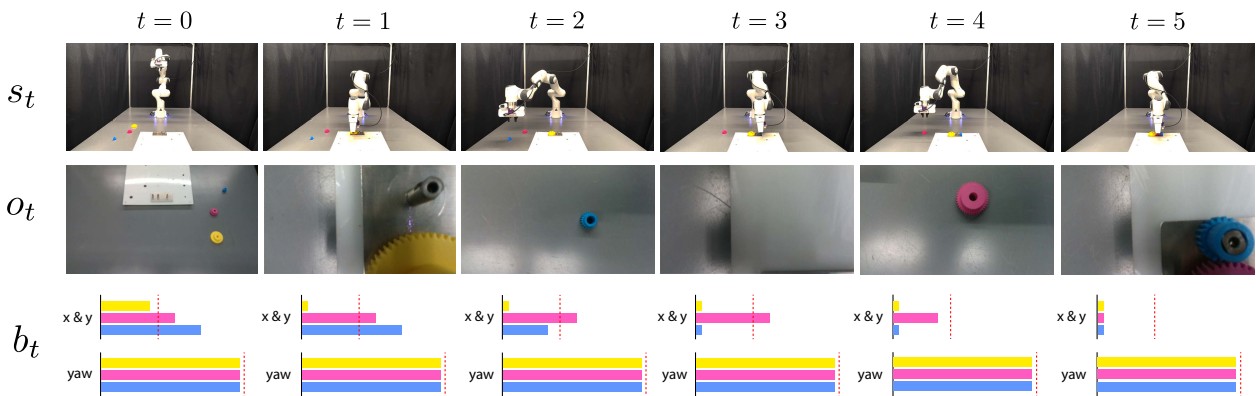

*Figure 4.* A multi-step manipulation plan using probabilistic pose estimation to estimate and update beliefs over time. The three rows show the robot state $s_t$, the observation $o_t$, and the robot belief $b_t$ at each timestep. The red dotted line in the belief indicates the marginal entropy thresholds for the x, y, and yaw (rotation around z) dimensions as determined by the learned normalizing flow. A belief with entropy surpassing the threshold line indicates the policy will likely fail. For full visualizations of the belief posteriors, flow distributions, and value maps, see Figure 14.

discussed in the following sections.

In addition to simulated experiments, we additionally evaluated the trained policies on a real-world gear insertion task. The results of those real-world experiments show that GoFlow results in more robust sim-to-real transfer as seen in Table 1 and in the supplementary videos.

# 6. Application to Multi-step manipulation

While reinforcement learning has proven to be a valuable technique for learning short-horizon skills in dynamic and contact-rich settings, it often struggles to generalize to more long-horizon and open ended problems (Sutton & Barto, 2018). The topic of sequencing short horizon skills in the context of a higher-level decision strategy has been of increasing interest to both the planning (Mishra et al., 2023) and reinforcement learning communities (Nasiriany et al., 2021). For this reason, we examine the utility of these learned sampling distributions as out-of-distribution detectors, or belief-space preconditions, in the context of a multi-step planning system.

## 6.1. Belief-space planning background

Belief-space planning is a framework for decision-making under uncertainty, where the agent maintains a probability distribution over possible states, known as the *belief state*. Instead of planning solely in the state space $\mathcal{S}$, the agent operates in the *belief space* $\mathcal{B}$, which consists of all possible probability distributions over $\mathcal{S}$. This approach is particularly useful in partially observable environments where there is uncertainty in environment parameters and where it

is important to take actions to gain information.

Rather than operating at the primitive action level, belief-space planners often make use of high-level actions $\mathcal{A}_\Pi$, sometimes called skills or options. In our case, these high-level actions will be a discrete set of pretrained RL policies. These high-level actions come with a *belief-space precondition* and a *belief-space effect*, both of which are subsets of the belief space $\mathcal{B}$ (Kaelbling & Lozano-Perez, 2013; Curtis et al., 2024). Specifically, a high-level action $\pi \in \mathcal{A}_\Pi$ is associated with two components: a precondition $\text{Pre}_\pi \subseteq \mathcal{B}$, representing the set of belief states from which the action can be applied, and an effect $\text{Eff}_\pi \subseteq \mathcal{B}$, representing the set of belief states that the action was designed or trained to achieve. If the precondition holds—that is, if the current belief state $b$ satisfies $b \in \text{Pre}_\pi$—then applying action $a$ will achieve the effect $\text{Eff}_\pi$ with probability at least $\eta \in [0, 1]$. More formally, if $b \in \text{Pre}_\pi$, then $\Pr\left(b_{t+1} \in \text{Eff}_\pi \mid b_t, \pi\right) \geq \eta$. Depending on the planner, $\eta$ may be set in advance, or calculated by the planner as a function of the belief.

This formalization allows planners to reason abstractly about the effects of high-level actions under uncertainty, which can result in generalizable decision-making in long-horizon problems that require active information gathering or risk-awareness.

## 6.2. Computing preconditions

In this section, we highlight the potential application of the learned sampling distribution $p_\phi$ and privileged value function $V_\psi$ as a useful artifacts for belief-space planning. In particular, we are interested in identifying belief-space

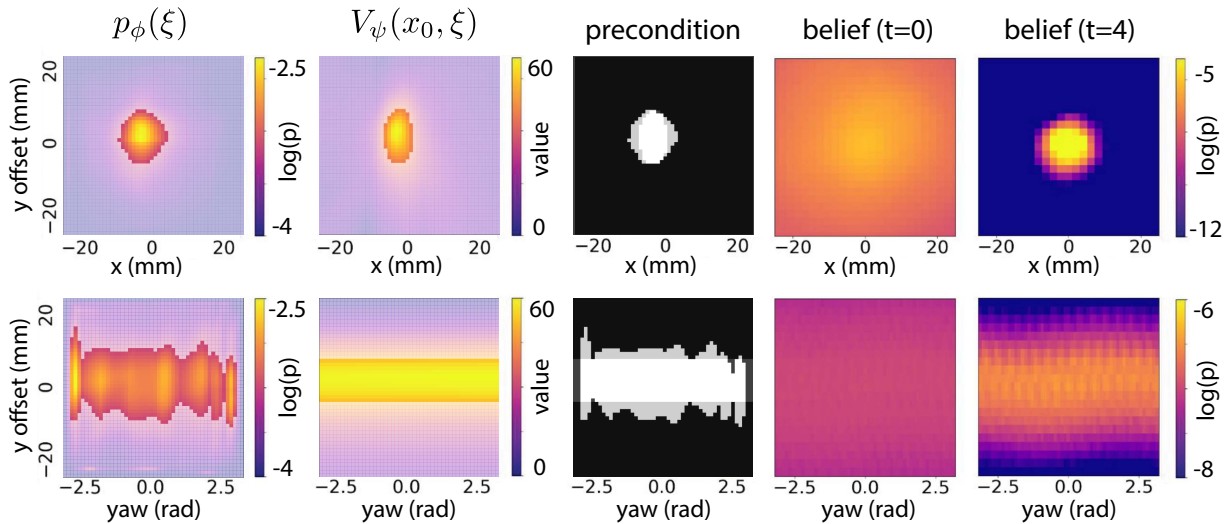

*Figure 5.* A visual example of the precondition computation described in Section 6.2 for the gear assembly plan shown in Figure 4. The two rows show two different projections of the 3D sampling space (x position vs y position in the top row and y position vs yaw rotation in the bottom row). We apply a threshold $\epsilon$ to the sampling distribution to remove low-probability regions (column 1). Additionally, we filter the value function by retaining only the regions where the expected value exceeds a predetermined threshold $\eta$ (column 2). The intersection of these two regions defines the belief-space precondition, indicating where the policy is likely to succeed (column 3). Comparing the precondition to the beliefs, we can see that the belief is not sufficiently contained within the precondition at $t = 0$ (column 4), but passes the success threshold $\eta$ at after closer inspection at $t = 4$ (column 5).

preconditions of a set of trained skills.

One point of leverage we have for this problem is the privileged value function $V_\psi(s, \xi)$, which was learned alongside the policy during training. One way to estimate the belief-space precondition is to simply find the set of belief states for which the expected value of the policy is larger than $J_T$ with probability greater than $\eta$ under the belief:

$$\text{Pre}_\pi = \left\{ b \in \mathcal{B} \mid \mathbb{E}_b \left[ \mathbb{1}_{V_\psi(s,\xi) > J_T} \right] > \eta \right\}. \quad (6)$$

However, a practical issue with this computation is that the value function is likely not calibrated in large portions of the state space that were not seen during policy training. To address this, we focus on regions of the environment where the agent has higher confidence due to sufficient sampling, i.e., where $p_\phi(\xi) > \epsilon$ for a threshold $\epsilon$. This enables us to integrate the value function over the belief distribution $b(x, \xi)$ and the trusted region within $\Xi$:

$$\text{Pre}_\pi = \left\{ b \in \mathcal{B} \mid \mathbb{E}_b \left[ \mathbb{1}_{V_\psi(s,\xi) > J_T} \cdot \mathbb{1}_{p_\phi(\xi) > \epsilon} \right] > \eta \right\} \quad (7)$$

.

Here, $\eta$ lower bounds the probability of achieving the desired effects $\text{Eff}_\pi$ (or value greater than $J_T$) after executing $\pi$ in any belief state in $\text{Pre}_\pi$. Figure 5 shows an example precondition for a single step of the assembly plan.

### 6.3. Updating beliefs

Updating the belief state requires a probabilistic state estimation system that outputs a posterior over the unobserved environment variables, rather than a single point estimate. We use a probabilistic object pose estimation framework called Bayes3D to infer posterior distributions over object pose (Gothoskar et al., 2023). For details on this, see Appendix A.4.1.

The benefit of this approach in contrast to traditional rendering-based pose estimation systems, such as those presented in (Wen et al., 2024) or (Labbé et al., 2022), is that pose estimates from Bayes3D indicate high uncertainty for distant, small, or occluded objects as well as uncertainty stemming from object symmetry. Figure 14 shows the pose beliefs across the multi-step plan.

### 6.4. A simple belief-space planner

While the problem of general-purpose multi-step planning in belief-space has been widely studied, in this paper we use a simple BFS belief-space planner to demonstrate the utility of the learned sampling distributions as belief-space preconditions. The full algorithm can be found in Algorithm 2.

An example plan can be seen in Figure 4. The goal is to assemble the gear box by inserting all three gears (yellow, pink, and blue) into the shafts on the gear plate. Each gear insertion is associated with a separate policy for each color

trained with GoFlow. In addition to the trained policies, the robot is given access to an object-parameterized inspection action which has no preconditions and whose effects are a reduced-variance pose estimate attained by moving the camera closer to the object. The robot is initially uncertain of the x, y, and yaw components of the 6-dof pose based on probabilistic pose estimates. Despite this uncertainty, the robot is confident enough in the pose of the largest and closest yellow gear to pick it up and insert it. In contrast, the blue and pink gears require further inspection to get a better pose estimate. Closer inspection reduces uncertainty along the x and y axis, but reveals no additional information about yaw dimension due to rotational symmetry. Despite an unknown yaw dimension, the robot is confident in the insertion because the flow $p_\phi$ indicates that success is invariant to the yaw dimension. This is due to the fact that success in the insertion task is defined by the distance between the bottom center of the gear and the base of the gear shaft, which is independent of gear rotation. For visualizations of the beliefs and flows at each step, see Appendix A.4.

## 7. Conclusion and discussion

In this paper, we introduced GoFlow, a novel approach to domain randomization that uses normalizing flows to dynamically adjust the sampling distribution during reinforcement learning. By combining actor-critic reinforcement learning with a learned neural sampling distribution, we enabled more flexible and expressive parameterization of environmental variables, leading to better generalization in complex tasks like contact-rich assembly. Our experiments demonstrated that GoFlow achieves higher coverage than traditional fixed and learning-based domain randomization techniques across a variety of simulated environments, particularly in scenarios where the domain has irregular dependencies between parameters. The method also showed promise in real-world robotic tasks including contact-rich assembly.

Moreover, we extended GoFlow to multi-step decision-making tasks, integrating it with belief-space planning to handle long-horizon problems under uncertainty. This extension enabled the use of learned sampling distributions and value functions as preconditions leading to active information gathering.

Although GoFlow enables more expressive sampling distributions, it also presents some new challenges. One limitation of our method is that it has higher variance due to occasional training instability of the flow. This instability can be alleviated by increasing $\beta$, but at the cost of reduced sample efficiency (see Appendix A.5). In addition, using the flow and value estimates for belief-space planning require manual selection and tuning of several thresholds which are environment specific. The $\eta$ parameter may be converted from a threshold into a cost in the belief space planner, which would remove one point of manual tuning. However, removing the $\epsilon$ parameter may prove more difficult, as it would require uncertainty quantification of the neural value function. Despite these challenges, we hope this work inspires further research on integrating short-horizon learned policies into broader planning frameworks, particularly in contexts involving uncertainty and partial observability.

## Acknowledgements

We gratefully acknowledge support from NSF grant 2214177; from AFOSR grant FA9550-22-1-0249; from ONR MURI grants N00014-22-1-2740 and N00014-24-1-2603; from the MIT Quest for Intelligence; and from the Robotics and Artificial Intelligence Institute. Aidan Curtis was supported by the NSF Graduate Research Fellowship. This work was partially conducted during a summer internship at Autodesk.

## Impact Statement

This paper presents work whose goal is to advance the field of Machine Learning. There are many potential societal consequences of our work, none which we feel must be specifically highlighted here.

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

# A. Appendix

## A.1. Code Release

The codebase for the project can be found here.

## A.2. Importance Sampling Proofs

Here we show how we obtain the importance-sampled estimates for both the reward term $\mathcal{R}$ and the entropy term $\hat{\mathcal{H}}$ in GoFlow(Algorithm 1). Specifically, we sample from a uniform distribution $u(\xi)$ over $\Xi$, rather than from the learned distribution $p_\phi(\xi)$ directly, in order to avoid collapse onto narrow regions of the parameter space.

### A.2.1. REWARD TERM: $\mathcal{R}$

We want an unbiased estimate of

$$\mathbb{E}_{\xi \sim p_\phi(\xi)}\big[ J_\xi(\pi) \big] = \int_\Xi p_\phi(\xi)\, J_\xi(\pi)\, \mathrm{d}\xi.$$

Let $u(\xi)$ be a uniform distribution over $\Xi$. Since $p_\phi(\xi)\, J_\xi(\pi) = \frac{p_\phi(\xi)}{u(\xi)}\, u(\xi)\, J_\xi(\pi)$, we can rewrite:

$$\int_\Xi p_\phi(\xi)\, J_\xi(\pi)\, \mathrm{d}\xi = \int_\Xi u(\xi)\, \frac{p_\phi(\xi)}{u(\xi)}\, J_\xi(\pi)\, \mathrm{d}\xi.$$

Hence, sampling $\xi_i \sim u(\xi)$ and averaging $\frac{p_\phi(\xi_i)}{u(\xi_i)} J_{\xi_i}(\pi)$ gives an unbiased Monte Carlo estimate of $\mathbb{E}_{p_\phi}\big[ J_\xi(\pi) \big]$.

If $\Xi$ has finite measure $|\Xi|$, then $u(\xi) = 1/|\Xi|$. Thus

$$\frac{p_\phi(\xi)}{u(\xi)} = p_\phi(\xi)\, |\Xi|.$$

Therefore, the empirical estimate becomes

$$\mathcal{R} = \frac{1}{B} \sum_{i=1}^{B} \big[ \big(p_\phi(\xi_i)\, |\Xi|\big)\, J_{\xi_i}(\pi) \big]$$

$$= \frac{|\Xi|}{B} \sum_{i=1}^{B} p_\phi(\xi_i)\, J_{\xi_i}(\pi).$$

where $\{\xi_i\}_{i=1}^{B} \sim u(\xi)$. This matches Line 7 in Algorithm 1.

### A.2.2. ENTROPY TERM: $\hat{\mathcal{H}}$

Similarly, to compute the differential entropy of $p_\phi(\xi)$, we have

$$\mathcal{H}(p_\phi) = -\int_\Xi p_\phi(\xi)\, \log p_\phi(\xi)\, \mathrm{d}\xi.$$

Again, we apply the same importance sampling trick via $u(\xi)$. We write:

$$p_\phi(\xi)\, \log p_\phi(\xi) = \frac{p_\phi(\xi)}{u(\xi)}\, u(\xi)\, \log p_\phi(\xi).$$

Hence

$$\int_\Xi p_\phi(\xi)\, \log p_\phi(\xi)\, \mathrm{d}\xi = \int_\Xi u(\xi)\, \frac{p_\phi(\xi)}{u(\xi)}\, \log p_\phi(\xi)\, \mathrm{d}\xi.$$

If $|\Xi|$ is the measure of $\Xi$ under $u(\xi)$, then $u(\xi) = 1/|\Xi|$. So the Monte Carlo estimate for $\int p_\phi(\xi)\, \log p_\phi(\xi)\, \mathrm{d}\xi$ becomes

$$\frac{|\Xi|}{B} \sum_{i=1}^{B} p_\phi(\xi_i)\, \log p_\phi(\xi_i),$$

with $\xi_i \sim u(\xi)$. Multiplying by $-1$ yields the differential entropy:

$$\mathcal{H}(p_\phi) = -\int p_\phi(\xi)\, \log p_\phi(\xi)\, \mathrm{d}\xi$$

$$\approx -\frac{|\Xi|}{B} \sum_{i=1}^{B} p_\phi(\xi_i)\, \log p_\phi(\xi_i),$$

which is exactly what we implement in Line 8 of Algorithm 1.

*Remark* A.1. Using uniform sampling $u(\xi)$ to approximate these terms provides global coverage of $\Xi$, helping prevent the learned distribution $p_\phi$ from collapsing around a small subset of parameter space. By contrast, if one sampled $\xi$ from $p_\phi(\xi)$ itself for these terms, the distribution might fail to expand to other promising regions once it becomes peaked.

## A.3. Domain Randomization Parameters

Below we describe the randomization ranges and parameter names for each environment. We also provide the reward success threshold ($J_T$) and cut the max duration of some environments in order to speed up training ($t_{max}$). $J_t$ was chosen to be below the optimal performance under no environment randomization. We verified that the trained policy still exhibited qualitatively successful performance at the target reward threshold. Lastly, we slightly modified the Quadruped environment to only take a fixed forward command rather than the goal-conditioned policy learned by default. Other than those changes, the first four simulated environments official IsaacLab implementation.

- **Cartpole parameters ($J_T = 50, t_{max} = 2s$):**
    - Pole mass: Min = 0.01, Max = 20.0
    - Cart mass: Min = 0.01, Max = 20.0
    - Slider-Cart Friction: Min Bound = 0.0, Max Bound = 1.0

- **Ant parameters ($J_T = 700, t_{max} = 2s$):**
    - Torso mass: Min = 0.01, Max = 20.0

- **Quadcopter parameters** ($J_T = 15, t_{max} = 2s$):

  – Quadcopter mass: Min = 0.01, Max = 20.0

- **Quadruped parameters** ($J_T = 1.5, t_{max} = 5s$):

  – Body mass: Min = 0.0, Max = 200.0
  – Left front hip friction: Min = 0.0, Max = 0.1
  – Left back hip friction: Min = 0.0, Max = 0.1
  – Right front hip friction: Min = 0.0, Max = 0.1
  – Right back hip friction: Min = 0.0, Max = 0.1

- **Humanoid parameters** ($J_T = 1000, t_{max} = 5s$):

  – Torso Mass: Min = 0.01, Max = 25.0
  – Head Mass: Min = 0.01, Max = 25.0
  – Left Hand Mass: Min = 0.01, Max = 30.0
  – Right Hand Mass: Min = 0.01, Max = 30.0

- **Gear parameters** ($J_T = 50, t_{max} = 4s$):

  – Grasp Pose x: Min = -0.05, Max = 0.05
  – Grasp Pose y: Min = -0.05, Max = 0.05
  – Grasp Pose yaw: Min = -0.393, Max = 0.393

## A.4. Multi-Step Planning Details

### A.4.1. UPDATING BELIEFS VIA PROBABILISTIC POSE ESTIMATION

Updating the belief state $b$ requires a probabilistic state estimation system that outputs a posterior over the state space $S$, rather than a single point estimate. Given a new observation $o$, we use a probabilistic object pose estimation framework (Bayes3D) to infer posterior distributions over object pose (Gothoskar et al., 2023).

The pose estimation system uses inference in an probabilistic generative graphics model with uniform priors on the translational $x$, $y$, and rotational yaw (or $r_x$) components of the 6-dof pose (since the object is assumed to be in flush contact with the table surface) and an image likelihood $P(o_{rgbd} \mid r_x, x, y)$. The object's geometry and color information is given by a mesh model. The image likelihood is computed by rendering a latent image $im^{rgbd}$ with the object pose corresponding to $(r_x, x, y)$ and calculating the per-pixel likelihood:

$$P(o_{rgbd} \mid r_x, x, y) \propto$$
$$\prod_{i,j \in C} \left[ p_{out} + (1 - p_{out}) \cdot P_{in}(o_{i,j}^{rgbd} \mid r_x, x, y) \right] \quad (8)$$

$$P_{in}(o_{i,j}^{rgbd} \mid r_x, x, y) \propto$$
$$\exp \left( -\frac{||o_{i,j}^{rgb} - im_{i,j}^{rgb}||_1}{b^{rgb}} - \frac{||o_{i,j}^{d} - im_{i,j}^{d}||_1}{b^{d}} \right) \quad (9)$$

where $i$ and $j$ are pixel row and column indices, $C$ is the set of valid pixels returned by the renderer, $b_{rgb}$ and $b_d$ are hyperparameters that control posterior sensitivity to the color and depth channels, and $p_{out}$ is the pixel outlier probability hyperparameter. For an observation $o_{rgbd}$, we can sample from $P(r_x, x, y \mid o_{rgbd}) \propto P(o_{rgbd} \mid r_x, x, y)$ to recover the object pose posterior with a tempering exponential factor $\alpha$ to encourage smoothness. We first find the maximum a posteriori (MAP) estimate of object pose using coarse-to-fine sequential Monte Carlo sampling (Del Moral et al., 2006) and then calculate a posterior approximation using a grid centered at the MAP estimate.

The benefit of this approach in contrast to traditional rendering-based pose estimation systems, such as those presented in (Wen et al., 2024) or (Labbé et al., 2022), is that our pose estimates indicate high uncertainty for distant, small, occluded, or non-visible objects as well dimensions along which the object is symmetric. A visualization of the pose beliefs at different points in the multi-step plan can be seen in Figure 14 in the Appendix.

## A.5. Hyperparameters

Below we list out the significant hyperparameters involved in each baseline method, and how we chose them based on our hyperparameter search. We run the same seed for each hyperparameter and pick the best performing hyperparameter as the representative for our larger quantitative experiments in figure 3. The full domain randomization (FullDR) and no domain randomization (NoDR) baselines have no hyperparameters.

### A.5.1. GOFLOW

We search over the following values of the $\alpha$ hyperparameter: $[0.1, 0.5, 1.0, 1.5, 2.0]$. We search over the following values of the $\beta$ hyperparameters $[0.0, 0.1, 0.5, 1.0, 2.0]$. Other hyperparameters include number of network updates per training epoch ($K = 100$), network learning rate ($\eta_\phi = 1e - 3$), and neural spline flow architecture hyperparameters such as network depth ($\ell = 3$), hidden features (64), and number of bins (8). We implement our flow using the Zuko normalizing flow library (Rozet et al., 2022).

### A.5.2. LSDR

Similary to GoFlow , we search over the following values of the $\alpha_L$ hyperparameter: $[0.1, 0.5, 1.0, 1.5, 2.0]$. Other hyperparameters include the number of updates per training epoch (T=100), and initial Gaussian parameters: $\mu = (\xi_{max} + \xi_{min})/2.0$ and $\Sigma = \text{diag}(\xi_{max} - \xi_{min}/10)$

| Method | FullDR | NoDR | DORAEMON | LSDR | ADR | GoFlow |
|---|---|---|---|---|---|---|
| Success Rate | 6/10 | 3/10 | 5/10 | 5/10 | 5/10 | **9/10** |

*Table 1.* Real-world experimental results with the statistically significant results bolded.

| | Cartpole | Ant | Quadcopter | Quadruped | Humanoid | Gears |
|---|---|---|---|---|---|---|
| FullDR | 0.060±0.021 | **0.014±0.006** | 0.000±0.000 | 0.012±0.007 | **0.195±0.044** | 0.000±0.000 |
| NoDR | 0.005±0.001 | 0.000±0.000 | 0.000±0.000 | 0.086±0.008 | 0.001±0.001 | **0.008±0.003** |
| DORAEMON | 0.015±0.005 | 0.000±0.000 | 0.000±0.000 | 0.029±0.008 | 0.000±0.000 | **0.012±0.004** |
| LSDR | 0.103±0.013 | 0.000±0.000 | **0.007±0.003** | 0.009±0.008 | **0.176±0.082** | 0.000±0.000 |
| ADR | 0.006±0.001 | 0.000±0.000 | 0.000±0.000 | 0.070±0.006 | 0.000±0.000 | **0.008±0.004** |
| GOFLOW | **0.138±0.005** | **0.056±0.033** | **0.010±0.006** | **0.275±0.019** | **0.274±0.076** | **0.035±0.019** |

*Table 2.* Mean and standard error (SDE) of the final reward value, with entries in bold indicating the best-performing method or methods not statistically significantly worse than the best (one-tailed z test, $\alpha = 0.05$).

### A.5.3. DORAEMON

We search over the following values of the $\epsilon_D$ hyperparameter: $[0.005, 0.01, 0.05, 0.1, 0.5]$. After fixing the best $\epsilon_D$ for each environment, we additionally search over the success thrshold $\alpha_D$: $[0.005, 0.01, 0.05, 0.1, 0.5]$.

### A.5.4. ADR

In ADR, we fix the upper threshold to be the success threshold $t^+ = J_T$ as was done in the original paper and search over the lower bound threshold $t^- = [0.1t^+, 0.25t^+, 0.5t^+, 0.75t^+, 0.9t^+]$. The value used in the original paper was $0.5t_H$. Other hyperparameters include the expansion/contraction rate, which we interpret to be a fixed fraction of the domain interval, $\Delta = 0.1 * [\xi_{max} - \xi_{min}]$, and boundary sampling probability $p_b = 0.5$.

### A.6. Coverage vs Range Experiments

We compare coverage vs. range scale in the ant domain. We adjust the parameter lower and upper bounds outlined in Appendix A.3 and see how the coverage responds to those changes during training. The parameter range is defined relative to a nominal midpoint $m$ set to the original domain parameters: $[m - (m - lower) * scale, m + (upper - m) * scale]$. The results of our experiment are shown in Figure 12

### A.7. Coverage vs. Threshold Experiments

In this experiment, we assess the sensitivity of coverage to the performance threshold $J_t$ for a single training seed. We experiment with threshold multipliers between $0$ and $1.2 \cdot J_t$. The results of this experiment can be seen in Figure 13. These results demonstrate that although coverage is highly

dependent on $J_t$, GoFlow largely dominates the coverage distribution for wide ranges of $J_t$.

### A.8. Real-world experiments

In addition to simulated experiments, we compare GoFlow against baselines on a real-world gear insertion task. In particular, we tested insertion of the pink medium gear over 10 trials for each baseline. To test this, we had the robot perform 10 successive pick/inserts of the pink gear into the middle shaft of the gear plate. Instead of randomizing the pose of the gear, we elected to fix the initial pose of the gear and the systematically perturb the end-effector pose by a random $\pm0.01$ meter translational offset along the x dimension during the pick. We expect some additional grasp pose noise due to position error during grasp and object shift during grasp. This led to a randomized in-hand offset while running the trained insertion policy. Our results show that GoFlow can indeed more robustly generalize to real-world robot settings under pose uncertainty.

### A.9. Statistical Tests

We performed a statistical analysis of the simulated results reported in Figure 3 and the real-world experiments in Table 1. For the simulated results, we recorded the final domain coverages across all seeds and performed pairwise t-tests between each method and the top-performing method. The final performance mean and standard deviation are reported in Table 2. Any methods that were not significantly different from the top performing method ($p < 0.05$) are bolded. This same method was used to test significance of the real-world results.

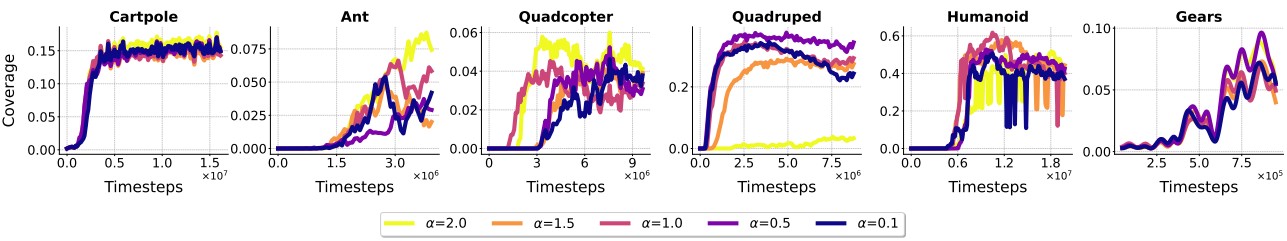

*Figure 6.* GoFlow hyperparameter sweep results for $\alpha$

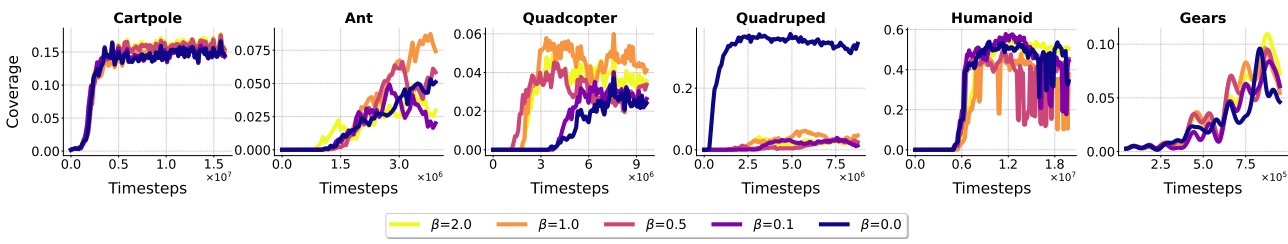

*Figure 7.* GoFlow hyperparameter sweep results for $\beta$ after fixing the best $\alpha$ for each environment

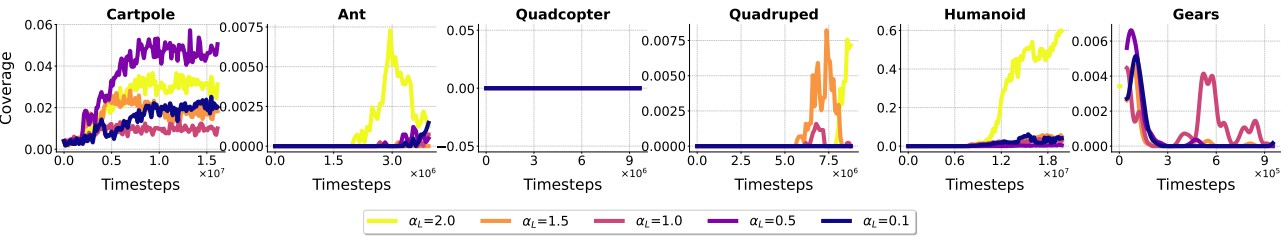

*Figure 8.* LSDR hyperparameter sweep results for $\alpha_L$

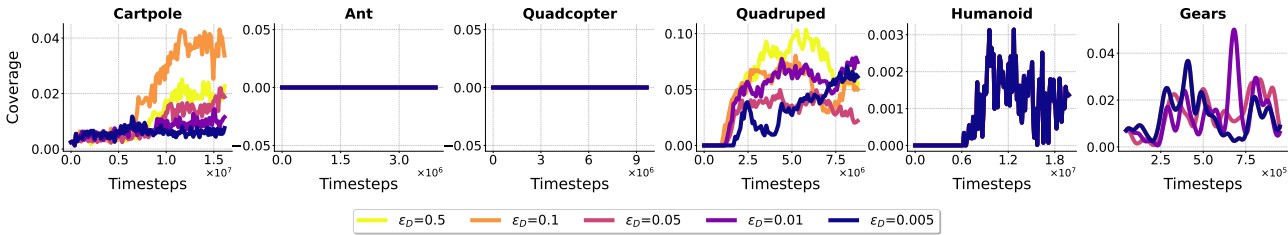

*Figure 9.* DORAEMON hyperparameter sweep results for $\epsilon_D$

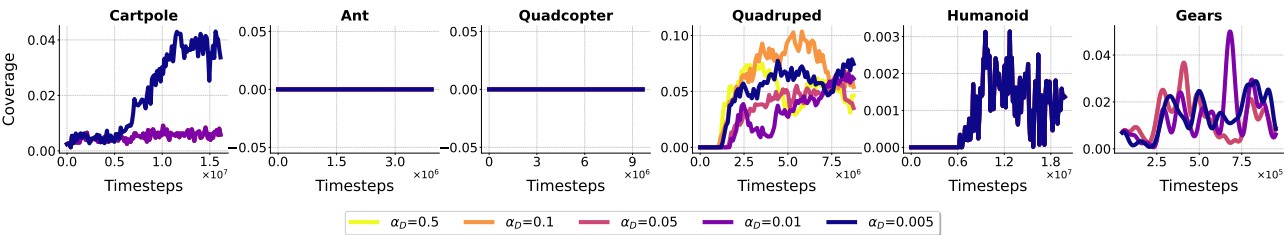

*Figure 10.* DORAEMON hyperparameter sweep results for $\alpha_D$ after fixing the best $\epsilon_D$

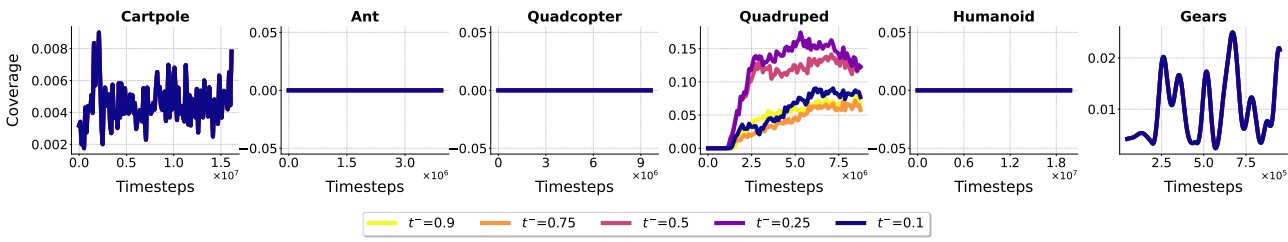

*Figure 11.* ADR hyperparameter sweep results for $t^-$

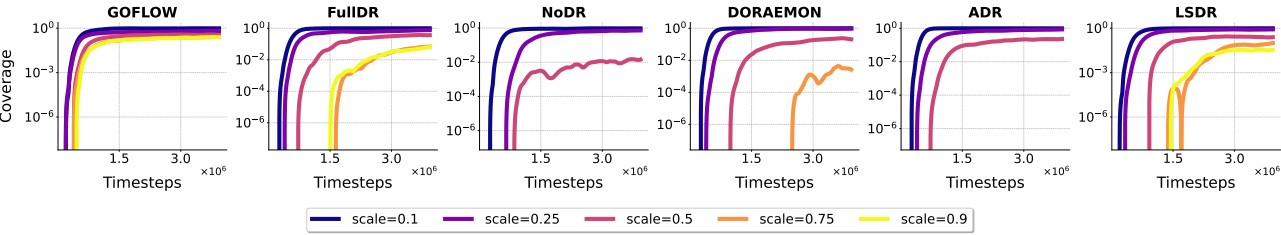

*Figure 12.* Coverage vs range experiment results discussed in Section A.6

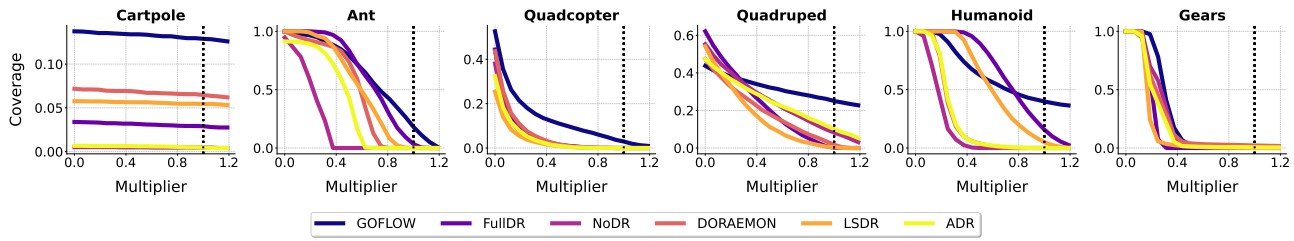

*Figure 13.* Coverage vs threshold experiment results discussed in Section A.7

|  | Cartpole | Ant | Quadcopter | Quadruped | Humanoid | Gears |
|---|---|---|---|---|---|---|
| FullDR | -920.132 | 362.454 | -2.286 | 0.108 | 657.414 | 8.627 |
| NoDR | -949.925 | 133.331 | -3.807 | -4.062 | 180.734 | 12.697 |
| DORAEMON | -930.417 | 86.742 | -1.767 | -2.472 | 235.516 | 11.378 |
| LSDR | -867.347 | 367.612 | -1.870 | -0.052 | 532.531 | 9.289 |
| ADR | -953.498 | 64.815 | -3.183 | -2.926 | 246.297 | 9.671 |
| GOFLOW | -820.510 | 198.302 | -1.000 | -2.814 | 376.652 | 14.209 |

*Table 3.* CVaR computed as the mean of the final rewards falling below the 10% percentile (VaR).

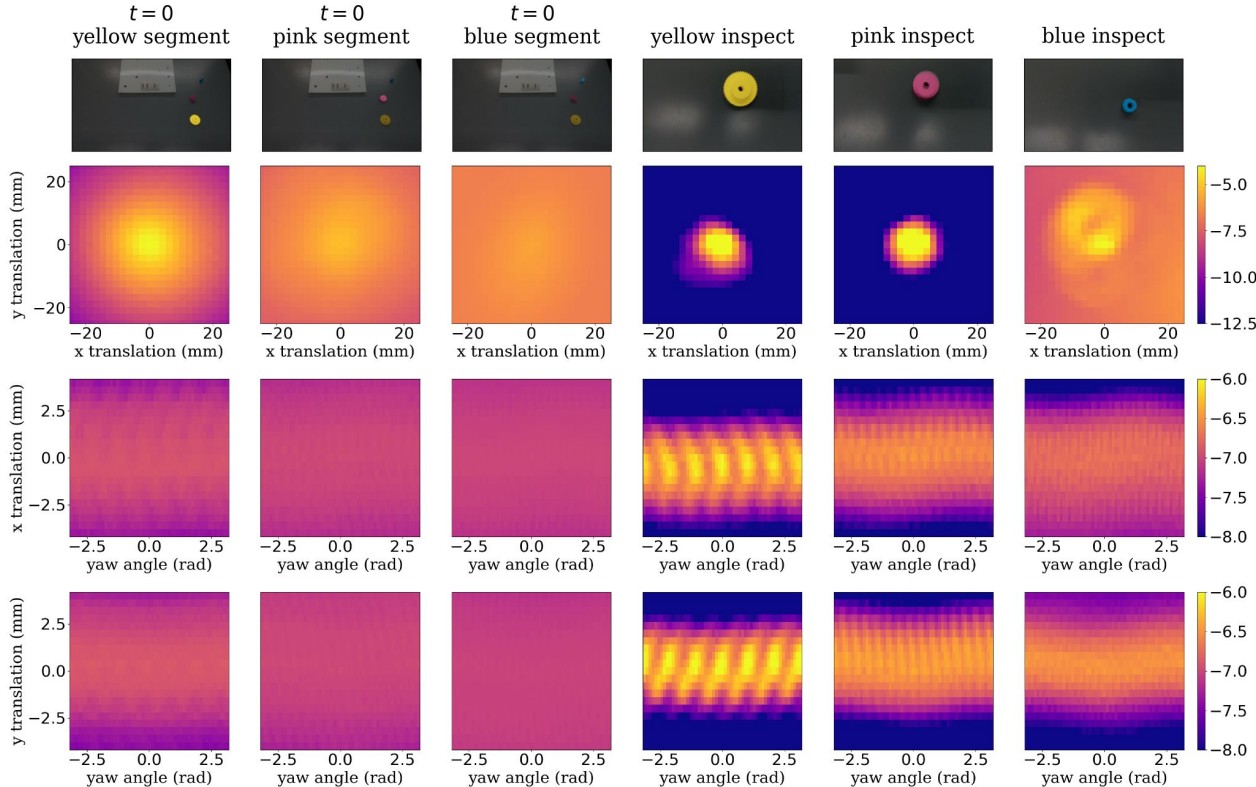

*Figure 14.* A visualization of the beliefs over the object pose under the initial image (first three columns) and after closer inspection (last three columns) as generated from the posterior of the model described in Section 6.3. The colormap corresponds to the log probability of the posterior pose estimate. All plots are centered around the most likely pose estimate under the image model.

---

**Algorithm 2** Belief-Space Planner Using BFS

---

**Require:** Initial belief state $b_0$, goal condition $G \subseteq \mathcal{B}$, set of skills $\mathcal{A}_\Pi$, success threshold $\eta$

 1: Initialize the frontier $\mathcal{F} \leftarrow \{b_0\}$
 2: Initialize the visited set $\mathcal{V} \leftarrow \emptyset$
 3: Initialize the plan dictionary Plan mapping belief states to sequences of skills
 4: **while** $\mathcal{F}$ is not empty **do**
 5:     Dequeue $b$ from $\mathcal{F}$
 6:     **if** $b \in G$ **then**
 7:         **return** Plan$[b]$                 $\triangleright$ Return the sequence of skills leading to $b$
 8:     **end if**
 9:     **for all** skills $\pi \in \mathcal{A}_\Pi$ **do**
10:         **if** $b \in \mathrm{Pre}_\pi$ given $\eta$ **then**
11:             $b' \leftarrow \texttt{sample}(\mathrm{Eff}_\pi)$
12:             **if** $b' \notin \mathcal{V}$ **then**
13:                 Add $b'$ to $\mathcal{F}$ and $\mathcal{V}$
14:                 Update Plan$[b'] \leftarrow$ Plan$[b] + [\pi]$
15:             **end if**
16:         **end if**
17:     **end for**
18: **end while**
19: **return Failure**                          $\triangleright$ No plan found

---

