# OpenReview forum: "Flow-based Domain Randomization for Learning and Sequencing Robotic Skills"
_ICML.cc/2025/Conference — ICML 2025 poster_

### Official Review · Reviewer_ejdi · 2025-03-10

**Overall Recommendation:** 3

**Summary:**

This work studies the domain randomization (DR) in reinforcement learning and the focus is on the design of task distribution. With the help of normalizing flows, the author proposes an entropy-regularized policy optimization methods for DR. The experiments are conducted in sim simulated and one real-world robotic domain.

---

I've updated the review based on the rebuttal.

**Claims And Evidence:**

It is worth noting the line number disappears in the complied pdf, which makes it difficult to point out the content. After going through the manuscript, I find some claims can be further polished and examples are attached as:

(1) **An ideal sampling distribution enables the policy to focus training on areas of the distribution that can feasibly be solved in order to maximize the overall success rate of the
trained policy while not wasting time on unsolvable regions of the domain.** on Page1. The claim does not hold all the time, for example, in curriculum learning, we tend to pay more time on solving unsolvable regions of the domain in the later learning process. The reference should be attached in the statement or writing should restrict the scope.

(2) **We show that GoFlow outperforms fixed and other learning-based solutions to domain randomization on a suite of simulated environments.** on Page2. There miss specific metrics or performance indicators about the “outperform” term.

(3) **Too broad a sampling distribution and the training focuses on unsolvable environments and falls into local minima.** On page3. What is the theoretical support or evidence support for this claim? My understanding about the broad sampling distribution is to cover scenarios as many as possible to achieve DR purpose.

Similar to the above cases, there are other claims, which can be polished in the future version.

**Essential References Not Discussed:**

[1] Tiboni, Gabriele, et al. "Domain randomization via entropy maximization." arXiv preprint arXiv:2311.01885 (2023).

[2] Ajay, Anurag, et al. "Distributionally adaptive meta reinforcement learning." Advances in Neural Information Processing Systems 35 (2022): 25856-25869.

[3] Wang, Cheems, et al. "Robust fast adaptation from adversarially explicit task distribution generation." arXiv preprint arXiv:2407.19523 (2024).

[4] Mehta, Bhairav, et al. "Active domain randomization." Conference on Robot Learning. PMLR, 2020.

**Experimental Designs Or Analyses:**

Overall, this work considers several scenarios on experimental design, however, more indicators and comparisons will strengthen this work.

**Methods And Evaluation Criteria:**

**I. Method:** Learning the task distribution for decision-making is a promising research direction and robust reinforcement learning is also a crucial consideration to bridge Sim2Real gap. The method of this work is the combination of the normalizing flow and DORAEMON [1].

However, there are some related works [2-4] that deserve discussion or comparison in experiments. For example, [2] considers the distribution shifts from a parameterized task distribution to increase RL robustness. [3] also uses normalizing flows to parameterize MDP distributions and learn and avoids the handcrafted task distribution design for robustness improvement. Though [2-3] are into meta RL, the optimization objectives also apply to DR cases. For [4], it considers the active domain randomization and also relates to robustness.


**II. Evaluation:** Throughout the manuscript, I find the coverage ratio works as the primary metric in terms of learning curves and table results. However, its concept is not well introduced at the beginning of evaluation. My understanding about the coverage is related to the reward threshold. Hence, some general metrics can be involved in strengthen this work.

(1) In terms of **generalization**, I recommend this work follows [1] and includes the learning curves about the success rate.

(2) In terms of **robustness**, I suggest some conditional value-at-risk (CVaR) of success rates for evaluation and OOD tasks performance.

Besides, I appreciate the author’s effort in real-world scenarios and this is plus of this work.


**Reference**

[1] Tiboni, Gabriele, et al. "Domain randomization via entropy maximization." arXiv preprint arXiv:2311.01885 (2023).

[2] Ajay, Anurag, et al. "Distributionally adaptive meta reinforcement learning." Advances in Neural Information Processing Systems 35 (2022): 25856-25869.

[3] Wang, Cheems, et al. "Robust fast adaptation from adversarially explicit task distribution generation." arXiv preprint arXiv:2407.19523 (2024).

[4] Mehta, Bhairav, et al. "Active domain randomization." Conference on Robot Learning. PMLR, 2020.

**Other Comments Or Suggestions:**

It would be great if the above suggestions can be incorporated in revision. Meanwhile, the method part needs to exclude some literature work and focuses more on the contributed points. After revision, this will be a strong paper.

**Other Strengths And Weaknesses:**

See the above

**Questions For Authors:**

No

**Relation To Broader Scientific Literature:**

Not applicable.

**Theoretical Claims:**

Not applicable. There is no theoretical claim in this work.

---

> ### Author Rebuttal · Authors · 2025-03-26
>
> Thank you for your detailed review and practical recommendations regarding our claims and experimental metrics, which have significantly strengthened our manuscript. Below we address each of your comments and questions.
>
> > … The claim does not hold all the time, for example, in curriculum learning, we tend to pay more time on solving unsolvable regions of the domain in the later learning process. The reference should be attached in the statement or writing should restrict the scope.
>
> We agree that this claim was imprecise, and there are other methods that train adversarially on difficult problems. While this would not be a good strategy in our class of problems due to the infeasibility of some parts of the parameter space. That statement has been updated to say "An alternative strategy is to learn an environment distribution during training with the aim of finding the broadest possible training distribution that can feasibly be solved in order to maximize the chances of transferring to an unknown target environment." Additionally, we added a section to related work discussing adversarial training.
>
> > There miss specific metrics or performance indicators about the “outperform” term.
>
> We have modified the statement to a more detailed claim about coverage.
>
> > Too broad a sampling distribution and the training focuses on unsolvable environments and falls into local minima. On page3. What is the theoretical support or evidence support for this claim? My understanding about the broad sampling distribution is to cover scenarios as many as possible to achieve DR purpose.
>
> The support for this claim comes from our experimental results and from the results of many other papers on learning for domain randomization. Figure 3 shows that full domain randomization (the broadest possible distribution) is insufficient for reaching the target success threshold. For example, in the “ant” domain where the goal is for the ant to run forward, full domain randomization learns a uniformly applied strategy of bracing the weight of the body so that the negative reward is not received from floor contact. The result is that none of the ants (even the lighter ones) can be considered to have a successful policy because none are running forward.
>
> > However, there are some related works [2-4] that deserve discussion or comparison in experiments.
>
> We agree that these are relevant papers, and we have updated our related work to include them.
>
> > Throughout the manuscript, I find the coverage ratio works as the primary metric in terms of learning curves and table results. However, its concept is not well introduced at the beginning of evaluation. My understanding about the coverage is related to the reward threshold. Hence, some general metrics can be involved in strengthen this work.
>
> Apologies for any confusion, we moved our discussion of the coverage metric to earlier in the evaluation section (section 5.2) and added detail to how it is calculated in our experiments.
>
> Regarding the other suggested metrics
> - Our coverage metric is the same as the success rate metric from Tiboni et al. More specifically, it is the proportion of environments from the entire parameter range that the policy is expected to succeed under. We prefer the name “coverage” because this quantity does not reflect the actual success rate on a robot.
> - We agree that this is a good metric to report. In our updated manuscript, we have added a new table with CVaR scores.  CVaR was computed as the mean of the final rewards falling below the 10\% percentile (VaR).

---

> > ### Comment · Reviewer_ejdi · 2025-04-04
> >
> > I thank the author's response. Some of my questions are addressed.
> >
> > In the updated manuscript, it is necessary to discuss the mentioned references [1-4] in detail or even include some in experimental comparison, where some of them learns the task distribution or uses normalizing flows.
> >
> > It also requires more efforts to polish the manuscript in both statements and background knowledge from scratch to improve its readability. Particular, remember to explain the evaluation and CVaR metrics at the beginning of experimental section.
> >
> > Also, the review score has been updated accordingly.

---

### Official Review · Reviewer_95Yv · 2025-03-11

**Overall Recommendation:** 3

**Summary:**

Domain randomization is a known and useful technique to transfer models trained in simulation to the richness of the real-world. There are several known methods in the past. The usual workflow of the algorithm is to learn or estimate the distribution of valid parameters that is solvable by the policy optimization simultaneously with the policy. It requires smart choices of loss function and training algorithms to achieve this goal. In this paper, the authors use a normalizing flow to fit the parameter distribution and use an entropy regularized loss function for training. The conjecture is that neural parameter distributions are more expressive and will be conducive to generalization. Results are demonstrated on some benchmarks to demonstrate more robust generalization as compared to previous baselines. Further, real-world demonstrations are shown on a manipulation task and extensions are made to multi-step planning.

**Claims And Evidence:**

The authors claim that using a neural normalizing flow for learning the distribution of domain parameters is an improvement on the state of the art for domain randomization. This is supported by adequate experimental evidence. Also, this same method is illustrated to be useful for learning the precondition distribution in a belief space planner.

**Essential References Not Discussed:**

The citations can be more comprehensive with more mention of domain adaptation, transfer learning and sim2real transfer. Some examples are mentioned below:

[A1] Yu, Wenhao, C. Karen Liu, and Greg Turk. "Policy transfer with strategy optimization." arXiv preprint arXiv:1810.05751 (2018).

[A2] Sagawa, Shogo, and Hideitsu Hino. "Gradual domain adaptation via normalizing flows." arXiv preprint arXiv:2206.11492 (2022).

Also, the authors are requested to expand further on these lines: "Some previous works have combined domain randomization with information gathering via system identification (Ramos et al., 2019; Sagawa & Hino, 2024)"

**Experimental Designs Or Analyses:**

There is a toy example which is useful for the reader to understand the essence of the paper. Then, there are results illustrated on MuJoCo benchmarks. The coverage metric is an indicator of how much of the parameter space results in a successful policy. Further experiments on a Bayesian multi-step planner illustrate a novel use case for the algorithm.

**Methods And Evaluation Criteria:**

The methods are sound and the evaluation criteria is acceptable. The proposed method achieves a bigger coverage range as compared to the baselines. The previous solution of fitting beta distributions has certain advantages. That is amenable to using a trust region constraint rather than a trust region regularizer which is adhoc.

**Other Comments Or Suggestions:**

“Reinforcement learning (RL) has proven to be a useful tool in robotics for learning control or action policies for tasks and systems which are highly variable and/or analytically intractable” - unclear statement

Fig 5 caption: “The thresholded sampling distribution is further thresholded by the value function to get the belief-space precondition” - unclear statement

Some additional information in the supplementary material could be discussed further in the main section of the paper. For example, there is a passing mention of the policy being independent to the yaw. This could be discussed in more detail.

**Other Strengths And Weaknesses:**

Strengths:

The flow of the paper with the motivation, method and experimental results are good. I find the illustrations on robot assembly tasks and belief-space planning a new addition. Overall, there is a knowledge gain for the reader.

The strength of the proposed method lies in the effectiveness despite its simplicity. More theoretical study on why this particular loss function and choice of normalizing flow is good will be interesting for future research.

Weaknesses:

The use of domain randomization is to result in domain generalization and sim2real. It is difficult to assume that domain randomization alone will achieve this. There could be some more discussion about the other approaches.

On the flipside, the methods by themselves are not very novel though effective. The entropy regularization objective, the normalizing flow and neural probablistic modeling are all well-known. They are used in an interesting and effective way.

The writing is unclear in certain places. See below.

**Questions For Authors:**

In terms of the choice of neural sampling distribution, LSDR also uses a learned distribution. Is the main difference that LSDR uses a neural network to fit the GMM parameters? Or do they learn the GMM parameters directly?

What happens if we increase the number of mixtures or the expressivity of the distribution without a neural network?

**Relation To Broader Scientific Literature:**

The topic of domain randomization is under discussion for some time and gained visibility by OpenAI's work on solving the sim2real problem for rubik's cube manipulation. Even though the methods are simple, the effectiveness is usually quite good. This paper is an useful extension. Apart from incremental advances on existing benchmarks and sim2real application, it introduces sampling distribution modeling in the context of belief state preconditions. Further, there is a lot of work in the use of domain randomization for out-of-distribution generalization in the context of images. The dynamics randomization is relatively less explored.

**Theoretical Claims:**

N/A

---

> ### Author Rebuttal · Authors · 2025-03-26
>
> Thank you for your constructive feedback and suggestions on expanding our related work and improving figure clarity, which have enhanced the overall presentation of our paper. Below we address each of your comments and questions.
>
> > The citations can be more comprehensive with more mention of domain adaptation, transfer learning and sim2real transfer. Some examples are mentioned below:
>
> We agree that these are relevant papers, and we have updated our related work to include them.
>
> > Also, the authors are requested to expand further on these lines: "Some previous works have combined domain randomization with information gathering via system identification (Ramos et al., 2019; Sagawa & Hino, 2024)"
>
> We have expanded on this discussion further in the related work section. That section of the related work now reads as follows:
> "Beyond training robust policies in simulation, learned sampling distributions can be tied to the real-world environmental conditions under which policies are likely to succeed. Previous works have integrated domain randomization with real-world interactions for more informed training distributions [citations] or to find the maximally effective real-world strategy [citations]. However, these methods often necessitate expensive policy retraining or data-intensive evolutionary search based on real-world feedback, posing challenges for real-time applications. Instead, we utilize our learned sampling distribution as an out-of-distribution detector within a multi-step planning framework, enabling fast and data-efficient information gathering in the real world."
>
> > The use of domain randomization is to result in domain generalization and sim2real. It is difficult to assume that domain randomization alone will achieve this. There could be some more discussion about the other approaches.
>
> We agree that domain randomization alone is not always sufficient for transfer. Hopefully our updated related work discussion helps highlight that real-world feedback is an important component of sim2real transfer.
>
> > In terms of the choice of neural sampling distribution, LSDR also uses a learned distribution. Is the main difference that LSDR uses a neural network to fit the GMM parameters? Or do they learn the GMM parameters directly?
>
> The LSDR baseline directly learns multivariate gaussian parameters. They don’t learn a GMM or use neural networks, but rely a simple Gaussian representation.
>
> > The writing is unclear in certain places. See below.
>
> Thank you for pointing these out. We have addressed all of these in the updated manuscript. Specifically, we made the following updates:
> 1. We clarified our statement regarding the usefulness of RL in the introduction. It now reads as follows: "Reinforcement learning (RL) is a powerful tool in robotics because it can be used to learn effective control policies for systems with highly complex dynamics that are difficult to model analytically. Unlike traditional control methods, which rely on precise mathematical models, RL learns directly from simulated or real-world experience [citations]."
> 2. We updated the caption of figure 5 to much more clearly state what each subfigure is. Please see our rebuttal to Reviewer aKt5 for details.
> 3. We elaborated on the yaw invariance property of the gear-insertion problem in the main text of the paper. It now reads as follows: "Despite an unknown yaw dimension, the robot is confident in the insertion because the flow $p_\phi$ indicates that success is invariant to the yaw dimension. This is due to the fact that success in the insertion task is defined by the distance between the bottom center of the gear and the base of the gear shaft, which is independent of gear rotation."

---

> > ### Comment · Reviewer_95Yv · 2025-04-07
> >
> > The authors have addressed most of my reviewer feedback. Most of the warranted fixes are minor and can be done easily.
> >
> > The presentation and writing in the first draft is effective. I am currently retaining my score.

---

### Official Review · Reviewer_CXok · 2025-03-13

**Overall Recommendation:** 3

**Summary:**

This paper proposes a normalizing flow based approach to learn sampling parameters for domain randomization. Instead of doing naive sampling for domain randomization hyper-params, the paper uses a more principled way (which has been proposed before). But different from previous works, this paper proposes using a normalizing flow based model to learn the sampling distribution and iteratively improve the policy by sampling domain parameters from this distribution. Experimental results in simple domains shows the effectiveness of the proposed approach. The proposed approach is also combined with a belief space planner to show how the proposed approach can be combined with traditional belief space planning to accomplish long horizon tasks.

**Claims And Evidence:**

Yes, the claims are well supported. The paper claims that their proposed domain randomization appraoch is more robust which is what the experiments correctly validate and show.

**Essential References Not Discussed:**

no

**Experimental Designs Or Analyses:**

Yes. The toy experiment is a bit contrived in my opinion since there is assumed to be this complex relation in the space. Clearly, a more complex model would be able to fit it much better than other naive approaches (most baselines).
For the other mujoco experiments it was not immediately clear how the base sampling distributions were chosen, but it seems they were chosen to higlight larger difference than baseline. Figure 12. in the Appendix provides a more clear picture. I guess one big concern would be how realistic are some of these assumptions made in the paper. In real-world scenarios we often do Sys-ID on the robot to find good initial set of hyperparams and then make it robust around that nominal set. But the proposed approach (in the main paper) uses a very broad initial set which is a bit more complex and clearly a non-learning based approach would not work in this scenario.

**Methods And Evaluation Criteria:**

Yes, although more complex tasks/benchmarks could be created (e.g. more complex control tasks). But even the current set of tasks seem adequate.

**Other Comments Or Suggestions:**

none

**Other Strengths And Weaknesses:**

The paper focuses on an important problem. It is quite well written and clearly explains the problem and the solution. The proposed approach is not extremely novel but is well executed.

I do have some concerns about the experiments. Basically, I am curious to hear how would the experiments (mujoco ones) change if the feasible domain regions are not that complex? Also, is the assumption made in the paper realistic?  Most real world robotics works such as OpenAI hand rubik's cube and recent legged robots work often use good old-fashioned engineering and robotics to solve similar problems. How challenging or complex would be to apply the proposed approach for such scenarios, since the base assumption is that a flow based model could learn complex sampling distributions.

Why was the choice of spline based normalizing flow used to learn the sampling distribution? That part does not seem to be motivated in any way (unless I missed it).

**Questions For Authors:**

please see above

**Relation To Broader Scientific Literature:**

Domain randomization (DR) is very important for making robots work in the real world especially sim2real. Most legged locomotion stuff relies on DR. However, in those scenarios we often do sys-ID and then use careful engineering/expertise to iterate on things. The proposed approach is more automated, however, it is unclear if it can be applied to complex real-world problems, i.e, does it obviate the need of engineering (most likely not). Overall, the problem statement is important,

**Theoretical Claims:**

No, there is minimal theory and the overall algorithm is sound.

---

> ### Author Rebuttal · Authors · 2025-03-26
>
> Thank you for your thoughtful analysis and positive review. Below we address each of your comments and questions.
>
> > For the other mujoco experiments it was not immediately clear how the base sampling distributions were chosen
>
> We attempted to select ranges for these parameters that were large enough to capture all possible physically realistic parameter settings.
>
> > I guess one big concern would be how realistic are some of these assumptions made in the paper. In real-world scenarios we often do Sys-ID on the robot to find good initial set of hyperparams and then make it robust around that nominal set. But the proposed approach (in the main paper) uses a very broad initial set which is a bit more complex and clearly a non-learning based approach would not work in this scenario.
>
> We agree with the assessment of how sim-to-real issues are typically addressed, and the limitations of this process are some of the driving motivations for this paper. First, this manual Sys-ID to find a nominal set of hyperparameters requires some human effort on a per-task basis, which we are able to eliminate by selecting broad enough ranges. Second, the nominal parameter values and resulting distributions may not be optimal ones for task performance and robustness. Lastly, the integrated planning system is much more efficient and effective with large parameter ranges as opposed to narrower distributions around nominal values. This is because less information gathering is required by policies that are robust to a larger space of parameters.
>
> > Basically, I am curious to hear how would the experiments (mujoco ones) change if the feasible domain regions are not that complex?
>
> If the feasible domain regions are centered and regularly shaped, we see the performance of other learning-based methods increase. The experiments in Appendix A.6 demonstrate this property.
>
> > Why was the choice of spline based normalizing flow used to learn the sampling distribution?
>
> We did not have a specific reason to use spline flows over other models outside of its superior performance compared to other architectures in the Zuko library.

---

### Official Review · Reviewer_aKt5 · 2025-03-14

**Overall Recommendation:** 2

**Summary:**

The paper proposes updating environment parameters for policy training by learning a sampling distribution parameterized as a normalizing flow. Normalizing flow is known to be capable of representing more expressive distributions. The distribution is trained to maximize policy performance, maximize its marginal entropy, and minimize the change from the previous distribution. The method is validated on a few simulated tasks showing improved coverage (percentage of the parameters where the policy performance is higher than some threshold) over the baselines, as well as being applied as a OOD detector for a real-world robot manipulation planner.

**Claims And Evidence:**

The paper's main claim is that the learned sampling distribution improves the overall performance in testing environments. The results are mostly shown in Fig. 3. While the results themselves look convincing, many details of the experiment setup seem missing (or should have been provided in the main text). For example, how are the initial distribution and target distribution decided? How are $\alpha$ and $\beta$ chosen in Fig. 3?

More critically, it was not clear how $J_t$, the threshold for reaching target performance at any environment, is chosen. I imagine the choice can largely affect the behavior of the curves shown in Fig. 3. More discussion/justification (or additional results on varying $J_t$) should be provided.

**Essential References Not Discussed:**

I think the line of work around task-driven system identification [1,2,3] is very relevant, also trying to identifying relevant parameters for improving policy performance.

[1] Muratore et al., Data-efficient domain randomization with bayesian optimization

[2] Ren et al., Adaptsim: Task-driven simulation adaptation for sim-to-real transfer

[3] Liang et al., Learning active task-oriented exploration policies for bridging the sim-to-real gap

**Experimental Designs Or Analyses:**

Again, I think the paper is missing important details on the experimental designs as discussed above.

**Methods And Evaluation Criteria:**

I think the set of tasks considered in the environments is fairly comprehensive, and the study on a real-world manipulation task is well-appreciated. However, I do feel the toy problem in Fig. 2 is too contrived given the nature of flow matching vs. the baselines.

**Other Comments Or Suggestions:**

I recommend putting GoFlow as either the first or the last method in the legend in Fig. 3.

**Other Strengths And Weaknesses:**

Some of the figures in the paper can be improved. I don't understand the $\theta$ part of Fig. 4. It also took me quite a while to understand Fig. 5, especially top row vs. bottom row. I think the caption can be vastly improved to provide more context and detailed explanations.

**Questions For Authors:**

Do you have intuition why quadruped works the best with $\beta=0$ as Fig. 7 shows?

**Relation To Broader Scientific Literature:**

The paper studies the effect of environment parameters for training effective control policies. Such study is fitting as we consider more generalizable policy. However, I do think the paper lacks more discussion on how the approach can be applied in more realistic real-world tasks (for improved performance instead of just for OOD detection).

**Theoretical Claims:**

The paper has a small amount of theoretical claims in Appendix A.2, and they look correct to me.

---

> ### Author Rebuttal · Authors · 2025-03-26
>
> Thank you for your detailed insights on experimental design and parameter selection, which have greatly helped us clarify our approach. Below we address each of your comments and questions.
>
> > … how are the initial distribution and target distribution decided? How are $\alpha$ and and $\beta$ chosen in Fig. 3?
>
> The target distribution is the same across all methods, as it is a uniform distribution over a set of physical properties such as link masses, joint frictions, and object poses. We attempted to select ranges for these parameters that were large enough to capture all possible physically realistic parameter settings.
> The initial distribution depends on the method. For GoFlow, the initial distribution is defined by the random initialization of the network.
> $\alpha$ and $\beta$ were chosen through a hyperparameter selection process detailed in Appendix A.5.
>
> > More critically, it was not clear how $J_t$, the threshold for reaching target performance at any environment, is chosen. More discussion/justification (or additional results on varying $J_t$) should be provided.
>
> $J_t$ was chosen to be slightly below the optimal performance under no environment randomization. We verified that the trained policy still exhibited “qualitatively successful” performance at the target reward threshold. For example, we verified that the ant still exhibits running behavior at the chosen target threshold. We have updated the manuscript to describe this selection process.
> Additionally, we performed an experiment showing how coverage changes with $J_t$. Although we cannot update the manuscript during the rebuttal process, new results in appendix section A.7 show that while the coverage is highly dependent on the choice of $J_t$, GoFlow outperforms baseline methods across almost all choices of $J_t$.
>
> > I think the line of work around task-driven system identification [1,2,3] is very relevant
>
> We agree that these are relevant papers, and we have updated our related work [1,2] and introduction [3] to include them.
>
> > I recommend putting GoFlow as either the first or the last method in the legend in Fig. 3.
>
> We have moved GoFlow to be the first method in Figure 3.
>
> > Do you have intuition why quadruped works the best with \beta=0 as Fig. 7 shows?
>
> While $\beta$ can help with training stability, it can also cause the distribution to converge more slowly. The quadruped domain specifically was less sensitive to large swings in the sampling distribution, and therefore did not benefit from larger $\beta$.
>
> > Some of the figures in the paper can be improved. I don't understand the \theta part of Fig. 4. It also took me quite a while to understand Fig. 5, especially top row vs. bottom row. I think the caption can be vastly improved to provide more context and detailed explanations.
>
> Thank you for this suggestion. We have expanded on the captions to figures 4 and 5 to improve clarity.
> For figure 4, we modified the figure to remove $\theta$ and replace it with yaw, which is how it was described in the caption and elsewhere in the paper. We also defined it in the caption.
> For Figure 5, we rewrote the entire caption to clearly describe the meaning of each column/row of subfigures. The new caption reads as follows:
>
> > "A visual example of the precondition computation described in Section 6.2 for the gear assembly plan shown in Figure 4. The two rows show two different projections of the 3D sampling space (x position vs y position in the top row and y position vs yaw rotation in the bottom row). We apply a threshold $\epsilon$ to the sampling distribution to remove low-probability regions (column 1). Additionally, we filter the value function by retaining only the regions where the expected value exceeds a predetermined threshold $\eta$ (column 2). The intersection of these two regions defines the belief-space precondition, indicating where the policy is likely to succeed (column 3).
>     Comparing the precondition to the beliefs, we can see that the belief is not sufficiently contained within the precondition at $t=0$ (column 4), but passes the success threshold $\eta$ at after closer inspection at $t=4$ (column 5)."

---

### Decision · Program_Chairs · 2025-05-01

**Decision:**

Accept (poster)

**Comment:**

The paper introduces GoFlow, a domain randomization in reinforcement learning that utilizes normalizing flows to learn the sampling distribution of environment parameters aiming to improve the robustness of learned policies by automatically discovering an effective sampling distribution through entropy-regularized reward maximization. The effectiveness of GoFlow is demonstrated on simulated tasks and real-world manipulation planning context.

Reviewers generally found the idea of using normalizing flows for domain randomization promising and the experimental results on the presented benchmarks convincing. The application to a real-world task and belief space planning was also seen as a positive contribution. However, reviewers raised concerns regarding missing experimental details, the contrived nature of the toy problem, the justification for certain design choices (like spline flows), and the limited scope of comparison with related works. Some reviewers also pointed out unclear writing and the need for more detailed explanations of metrics.

In response, the authors clarified experimental setups, hyperparameter selection, and the motivation behind their approach. They agreed to include missing related works and improve the clarity of the manuscript, including figure captions and explanations of the coverage metric. While reviewers acknowledged the rebuttals and some updated their scores, some minor reservations persisted regarding the novelty.

The potential of contribution of automatically learning domain randomization distributions using normalizing flows is exciting, and the results, including real world robot manipulation, are positive. The remaining concerns are relatively minor.